# Non-Fatal Drowning Risk Prediction Based on Stacking Ensemble Algorithm

**DOI:** 10.3390/children9091383

**Published:** 2022-09-14

**Authors:** Xinshan Xie, Zhixing Li, Haofeng Xu, Dandan Peng, Lihua Yin, Ruilin Meng, Wei Wu, Wenjun Ma, Qingsong Chen

**Affiliations:** 1School of Public Health, Guangdong Pharmaceutical University, Guangzhou 510200, China; 2Guangdong Provincial Institute of Public Health, Guangdong Provincial Center for Disease Control and Prevention, Guangzhou 511430, China; 3Department of Public Health, School of Medicine, Jinan University, Guangzhou 510630, China; 4Guangdong Provincial Center for Disease Control and Prevention, Guangzhou 511430, China

**Keywords:** drowning, machine learning, stacking ensemble, risk-factors, prediction

## Abstract

Drowning is a major public health problem and a leading cause of death in children living in developing countries. We seek better machine learning (ML) algorithms to provide a novel risk-assessment insight on non-fatal drowning prediction. The data on non-fatal drowning were collected in Qingyuan city, Guangdong Province, China. We developed four ML models to predict the non-fatal drowning risk, including a logistic regression model (LR), random forest model (RF), support vector machine model (SVM), and stacking-based model, on three primary learners (LR, RF, SVM). The area under the curve (AUC), F1 value, accuracy, sensitivity, and specificity were calculated to evaluate the predictive ability of the different learning algorithms. This study included a total of 8390 children. Of those, 12.07% (1013) had experienced non-fatal drowning. We found the following risk factors are closely associated with the risk of non-fatal drowning: the frequency of swimming in open water, distance between the school and the surrounding open waters, swimming skills, personality (introvert) and relationality with family members. Compared to the other three base models, the stacking generalization model achieved a superior performance in the non-fatal drowning dataset (AUC = 0.741, sensitivity = 0.625, F1 value = 0.359, accuracy = 0.739 and specificity = 0.754). This study indicates that applying stacking ensemble algorithms in the non-fatal drowning dataset may outperform other ML models.

## 1. Introduction

The World Health Organization (WHO) reported that drowning is a serious public health challenge, causing 236,000 deaths annually worldwide [1]. Drowning is the fourth-leading cause of unintentional injury death across all injury types, accounting for 7% of all injury-related deaths [2]. Over 20% of drowning deaths in the Western Pacific Region occurs among children below the age of 15, exceeding the sum of the number of deaths due to HIV/AIDS, meningitis, malaria, dengue, malnutrition, respiratory diseases and hepatitis [3]. Drowning is the leading cause of injury-related death in children from age 1 to 14 years globally [4]. In China, the burden of disease attributed to drowning in children is significant. The highest drowning rates are among children aged 1–4 years (4.6/100,000), followed by children aged 10–14 years (3.54/10,000), 15–19 years (3.26/100,000) and 5–9 years (2.66/100,000) in 2019 [5]. The drowning-associated mortality rate in China reached 3.18 per 100,000 people in 2019, which is much higher than in the developed counties [5,6]. Presently, drowning remains a major threat to children’s lives in China and requires urgent action.

Previous studies have found that the risk factors of drowning are various, including the age, gender, educational level, usage of alcohol, etc. [7,8,9,10]. One study reported that the drowning-associated mortality rate was higher in males than in females, with a ratio of 5.0:2.1 [4]. A case-controlled study in China found that a lack of experience of playing in water and lack of parental close supervision were risk factors for drowning in children aged 5–14 years [11]. When encountering open water, children are more likely to have adverse outcomes due to their lack of strength, poor swimming skills and lack of water survival skills, etc. [12,13,14]. The previous studies have primarily focused on the risk factors of non-fatal drowning; only a few studies have paid attention to developing a better machine learning (ML) model [15]. An excellent ML model may provide a novel risk-assessment insight on the prediction of non-fatal drowning in children, potentially leading to its prevention.

The usage of machine learning algorithms has exponentially increased in recent years due to their strong predictive ability. ML algorithms are currently used in a variety of fields including medicine, economics and environmental sciences. [16,17,18]. ML algorithms have been shown to be successful in predicting the risk of cardiac mortality, as reported in a study from America [18]. A study in South Korea found that ML can improve the prediction of long-term outcomes in ischemic stroke patients [19]. Studies about the use of ML algorithms in the prediction of drowning are rare, and most previous studies implemented single learning algorithms. A prior study in China developed a logistic regression model to predict the risk of drowning among children, based on the risk factors found in non-fatal drowning [16]. However, the predictive ability of this study is low, because the authors did not work with imbalanced datasets and only used a single learning approach. Stacking is an algorithm that takes the outputs of base learners as input data and combines the input data to achieve a better output prediction. To increase the model performance of the non-fatal drowning prediction, we developed a stacking ensemble algorithm.

Drowning is a process during which breathing is impaired by submersion/immersion in a liquid. It includes non-fatal and fatal drowning [20]. When the process of drowning is interrupted by a successful rescue, it is considered to be a non-fatal drowning [21]. We applied survey data in Qingyuan City, China on non-fatal drowning in children to compare different machine learning algorithms and developed a stacking ensemble model. Our finding may provide a novel risk-assessment insight on drowning prevention, which may act as a supporting tool for the prediction of non-fatal drowning in children.

## 2. Methods

### 2.1. Study Site

The data used in this study were derived from a survey conducted in the two townships of Qingyuan City, a 19,152 square kilometer area in northern Guangdong Province, China. The area is characterized by many rivers and ponds. All students aged 8 to 18 years in grades 3 to 8 were selected, using multiple-stage cluster random sampling. Finally, a total of 8390 students were included in this study. We have provided the survey questionnaire (in English) as a Appendix A.

### 2.2. Data Collection

We conducted a cross-sectional survey among 8390 students in November 2013. A questionnaire was distributed to investigate the non-fatal drownings among students in Qingyuan City. The questionnaire was designed based on our previous studies of drowning prevention. All the participants were asked to complete the questionnaire in the classroom under the guidance of well-trained investigators. The survey included general information (such as age, gender, grade), and drowning-related information such as the concept of drowning (drowning can be prevented or not, the function of flotation devices), drowning-related risk factors (water area on the way to school, swimming skill), high-risk behaviors (such as swimming in the pond, diving into open water) and the student’s experience of drowning (prior experience of drowning or not). All of the questionnaires were checked by the investigators upon completion. All the data were found to be complete.

### 2.3. Statistical Analysis

#### Data Pre-Processing and Variables Selection

Unbalanced datasets are types of data in which the number of observations in one or more subclasses is much larger than in the others [22]. Imbalanced datasets are those in which a severe skew in the class distribution influences the performance of the machine learning algorithms, leading to a selective bias against the minority class. Applying a resample method to achieve a balanced dataset is an effective means to correct an imbalanced dataset [23]. Adding 1:7.28 (1013 vs. 7377) participants in the minority class (children who have experienced drowning) to the majority class (children who have never experienced drowning), we applied a replicated sample method to delete examples from the majority class with a ratio of 1:1, called under-sampling. The dataset was subsequently randomly divided into training data and testing data, with a ratio of 7:3 and sample sizes of 5824 and 2496, respectively. The training groups were used to select the optimal hyperparameter and the testing groups were used to evaluate the performance of the model.

The feature selection is an essential step in all machine learning. We have used two methods for the variable selection including the chi-square test and variable importance scores. Our dataset contained 20 variables, which can be found in the Appendix A. Finally, we determined the nine most important variables and obtained new, optimum non-fatal drowning subsets by applying two feature selection techniques. In the study, the nine important variables were selected for the development of the model.

### 2.4. Model Development

#### 2.4.1. Develop Base Learners

The stacking ensemble contains several base learners and secondary learners [24]. The algorithms used for the stacking ensemble in this study consisted of three models. We used a two-stage approach to fit the models in our study. Generally, base learners should be both accurate and diverse [24]. In the first stage, the base learners were generated from the training data by several base learning algorithms, including logistic regression (direction = forward), random forest (ntree = 1000, mtry = 8), and support vector machine (the kernel functions as the polynomial kernel, cost = 4, and coef = 5.). The secondary learner was then selected, based on the principle of keeping the algorithm simple in order to avoid over-fitting [25]. In the second stage, logistic regression was selected as the second-level learning algorithm. The settings of its parameters were obtained through five-fold cross-validation.

#### 2.4.2. Develop a Stacking Ensemble Model

As shown in Figure 1 and Figure 2, the process is accomplished by three steps, including dividing the data set, training the base learner, and training and evaluating the ensemble learner. In the first step, we divided the original data set randomly into the training data and the testing data, with a ratio of 7:3. The stacking ensemble was based on five-fold cross-validation. For example, the original training data was initially split into five folds randomly. During the first iteration, the first fold was used as the validation set, and the rest were delegated to the training set. The training set was then used to train the three base learners, respectively, and the validation set was used to test the ML models and to calculate the predictive value (y1^). The original test set was used to test the three base learners and generate the outcome value (y1*^). Next, we repeated this process, but using the different folds, and generated a new training set (y1^, y2^, y3^, y4^, y5^) and a new testing set (sum (y1*^, y2*^, y3*^, y4*^, y5*^)/5). Lastly, the new training set was applied to the second learner (LR) and used to calculate the eventual prediction outcome through the new testing data from the base learners.

### 2.5. Modeling Evaluation

We evaluated all the learning algorithms on the testing data set by the AUC (Area Under Curve), F1-value, accuracy (ACC), sensitivity (SE), and specificity (SP). We used the confusion matrix to calculate these indicators. The calculation method is:SE = TP/(TP + FN)SP = TN/(FP + TN)ACC = (TP + TN)/(TP + FN + FP + TN)F-value = 2 × TP/(2TP + FP + FN)

The TP, TN, FP and FN represent the number of true positives, true negatives, false negatives and false positives, respectively. Taking SN as the ordinate and SP as the abscissa to construct the ROC curve and the AUC value is equal to the area under the ROC curve.

All the statistical analyses in this study were processed by R software (R Core Team, https://www.R-project.org/ (accessed on 15 June 2022), version 4.0.2, R Foundation for Statistical Computing). We used the “stats” and “randomForest” packages, respectively, to conduct the logistic and random forest models, and the “e1071” package to perform the support vector machine model.

## 3. Results

### 3.1. Univariate Variables Selection

The univariate feature selection method uses the chi-squared (χ^2^) test for the non-fatal drowning attributes to select the 10 best variables in our dataset. We removed the variables that are not statistically significant, including the number of siblings, family ranking, whether open water is well protected near the home or school, and the distance from their home to any open water. Table 1 describes the distribution of the non-fatal drownings among the different subgroups. A total of 8320 children with 4369 males and 3951 females were included, and approximately 12.2% (1013/8320) had experienced non-fatal drowning during the past year. The non-fatal drowning ratio of students in grades 3 to 6 (75.81%, 768/1013)) and males (64.07%, 649/1013) who had ever experienced non-fatal drowning are higher than their counterparts in grades 7–8(24.19%, 245/1013) and the females (35.93%,364/1013). The introverts are found to be more likely to suffer non-fatal drowning than the other students (χ^2^ = 32.37, *p* < 0.01). The students with self-reported better swimming skills are also more likely to experience non-fatal drowning.

### 3.2. Feature Important

Figure 3 shows the relative importance of the factors associated with non-fatal drowning. The nine critical factors were the frequency of swimming in open water, the distance between the school and the surrounding open waters, the swimming skills, the personality (introvert), relationship with family members, relationship with classmates, sex and grade. Finally, the importance of each variable for the prediction of non-fatal drowning is evaluated, along with the ranking each factor. In our study, the frequency of swimming in open water is the most important variable.

### 3.3. Evaluation of Various Models

Table 2 and Figure 4 show the prediction performance of each algorithm. The areas under the curve (AUC) of the different models are 0.741 (stacking), 0.736 (LR), 0.717 (SVM), and 0.705 (RF), respectively. Compared with the other base models, the stacking generalization achieved the best performance in the non-fatal drowning prediction (AUC = 0.741, sensitivity = 0.625, F1 value = 0.359, accuracy = 0.739 and specificity = 0.754).

## 4. Discussion

Drowning is the leading cause of death among children in many countries, which leads to a great mortality burden [26]. This study, for the first time, identified the high-risk features for non-fatal drowning, based on the feature-importance rank and statistical analysis. We were then able to build various non-fatal drowning prediction models, based on different algorithms, and to compare the performance of each algorithm. Finally, this study indicates that applying stacking ensemble algorithms in the non-fatal drowning datasets may outperform the other single ML models. Our findings may provide a novel risk-assessment insight on non-fatal drowning prevention and may serve as a supporting tool for the prediction of non-fatal drowning.

In general, the risk factors associated with drowning are multi-dimensional, including the individual characteristics, natural environment and socio-economic factors [12,13,15]. In this study, we found that the frequency of swimming in open water, distance from open water around the school, swimming skills, personality (introvert), etc., are most closely correlated with children drowning. Previous studies also found that swimming in natural water bodies without adult supervision and playing in or around natural waters are critical risk factors for non-fatal drowning among children [27]. Natural water bodies are the most common sites of children drowning [27]. In rural districts of low- and middle-income regions, children are easily exposed to natural waters, which may not be surrounded by protective barriers [28]. Compared with adults, children are at higher risk because of weaker risk-awareness, a lack of physical strength, and inadequate swimming and water survival skills [12,13]. Self-reported swimming skills were shown to be associated with drowning in this study. In rural areas, children with better swimming skills might swim more frequently and engage in risk-taking behaviors while in the water, which may have contributed to our results. There is no doubt that poor swimmers are certainly more likely to drown when wading than those who swim well.

The Chinese government has made great efforts to implement drowning-prevention programs by successively targeting pre-school, primary, and middle school students, since 2006. Several studies have shown that these programs have been successful in reducing the number of children who drown [10,28]. A study conducted by the Chinese Center for Disease Control and Prevention revealed a decreasing trend in drowning mortality among children under five years of age in China, from 10.43 to 5.22 per 100,000 persons, between 2006 and 2017 [28]. The consensus is that children’s drowning is preventable through improved education, engineering, legislation and the enforcement of new policies. Education for children including basic swimming and water safety skills have proven to be effective in children’s drowning prevention [10]. Education for parents is also believed to be critical. A drowning study in Bangladesh found that nearly 80% of drowning among children occurred within 20 m of their home [13]. As parents become increasingly aware of where their children could drown, they will more diligently supervise their children. Installing barriers controlling the access to water is also a proven effective intervention in the prevention of drowning [29]. A case-controlled study in Mexico found that the risk of drowning for children with a well at home was almost seven-times that of children without a water well [30]. Although several efforts have been made to prevent children from drowning, the drowning mortality rate in China remains high [10]. Therefore, exploring new technologies persistently and combining them with the original prevention interventions is necessary.

The classification of learning algorithms aims to build a classifier from a training dataset in order to accurately predict testing samples [31]. The performance of the prediction models can be evaluated using different metrics, such as sensitivity, specificity, accuracy and the area under the ROC curve (AUC) [32]. Studies show that the AUC is more statistically discriminant than other methods when evaluating the predictive ability of the classification algorithm [31]. In this study, we found the performance of stacking is better than other single algorithms, which is consistent with previous studies [33,34]. A study in China found that, compared to other classification methods, a stacking ensemble showed superiority in processing different types of cancer data [33]. A Korean study also found a stacking ensemble achieved a high performance in breast cancer prediction [34]. The stacking ensemble is currently widely used in many fields because of its strong flexibility and scalability in ensemble learning algorithms [33]. Stacking ensemble algorithms combine multiple primary learners to achieve higher generation performance.

In this article, the drowning datasets after the features were selected are used as the input to develop and evaluate the models. To test the quality of the proposed method, we compared this method with logistic regression, random forest, support vector machine, and LR stacking. Among these methods, the AUC of all models is greater than 0.7, which proves that these models have a good generalization performance without overfitting [24].

As shown in Table 2 in the base learners, the random forest method obtained the best sensitivity, with 0.667. The logistic regression method achieved the highest accuracy of 0.740 and specificity of 0.758. In addition, the AUC and sensitivity of the support vector machine are the lowest. In the secondary learner, the stacking ensemble learning methods achieved an AUC of 0.741, a sensitivity of 0.625, an F1-value of 0.359, an accuracy of 0.739, and a specificity of 0.754. Compared with logistic regression, the largest increases in the results were observed in the SE (+20%), AUC (+5%) and F1 values (+7%), when compared with the logistic regression, random forest, and support vector machine models. The overall performance of the stacking strategy was superior to the other models. Based on our assessment indicators, logistic regression has better accuracy and can make correct judgments but is less sensitive. The random forest has a high sensitivity and identifies positive samples easily, but its accuracy is poor. The support vector machines have a perfect mathematical theory and some accuracy. By combining the advantages of these three single models, we have achieved the stacking ensemble model. The stacking fit is the best and has the highest AUC, sensitivity and F1 values.

This study has some strengths. First, our study was one of the few studies which established drowning risk prediction models for children. Moreover, we built various models based on different algorithms and compared their prediction abilities. Several limitations should also be acknowledged. Our study population was conducted on children and does not represent all ages. The data in this study were also collected from self-reporting done one year ago, which may induce recall bias. Steps to improve the data collected on non-fatal drowning and establish a national monitoring system are warranted. Concerning our methods, we only used three base learners; perhaps a combination of more learners could have produced better predictions. At the same time, the result of this research could be a valuable reference for comparing traditional integrated learning algorithm bagging and boosting.

## 5. Conclusions

Our study shows that the frequency of swimming in open water, the distance between the school and surrounding open waters, swimming skills, personality (introvert), relationship with family members, relationship with classmates, sex and grade are closely correlated with children experiencing non-fatal drowning. The stacking ensemble achieved better performance than the SVM, RF and LR in predicting non-fatal drowning in children. Our finding may provide a novel risk-assessment insight on non-fatal drowning prevention, which may act as a supporting tool for the prediction of non-fatal drowning. The result of this research could be a valuable reference for choosing machine learning models for further non-fatal drowning research. It is possible that effective approaches could be promoted in the future.

## Figures and Tables

**Figure 1 children-09-01383-f001:**
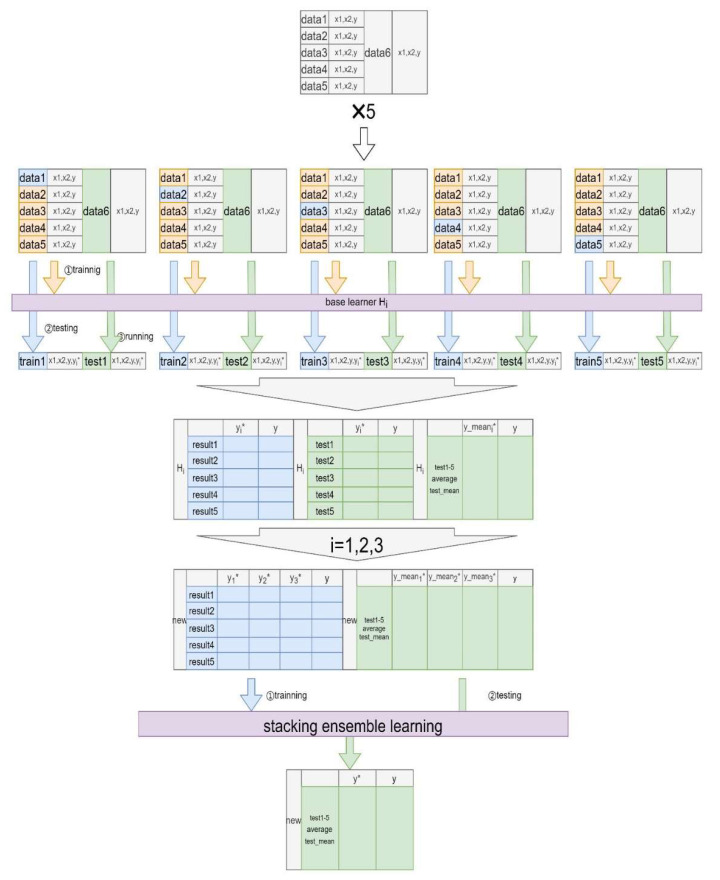
Drowning prediction model based on the stacking ensemble learning. Note that the original dataset contains independent variables (x1, x2, etc.) and a dependent variable (y). In the new, stacking dataset, yi* (i = 1,2,3) and y_meani* (i = 1,2,3) represent the prediction probabilities of the three base algorithms on the original training and testing sets, respectively.

**Figure 2 children-09-01383-f002:**
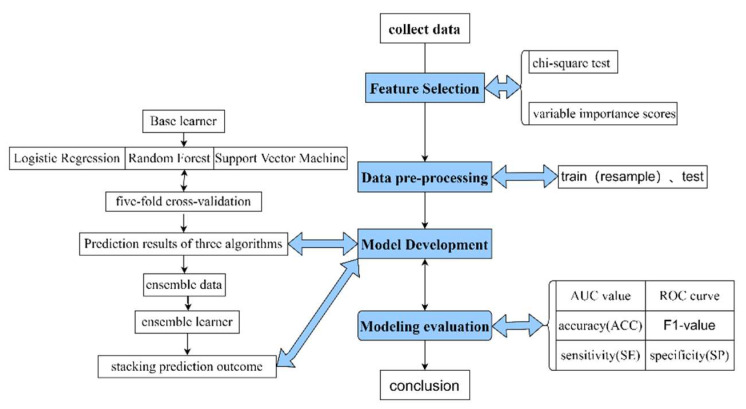
The experimental flow of the research method.

**Figure 3 children-09-01383-f003:**
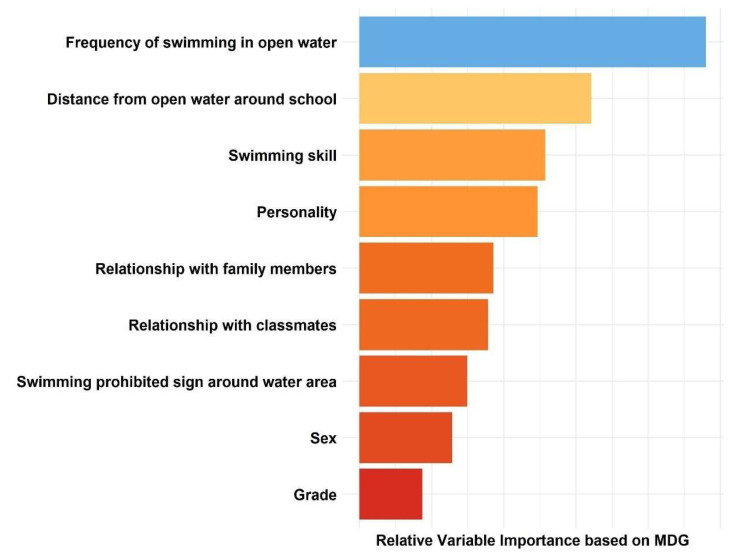
The relative importance of risk factors using a random forest model.

**Figure 4 children-09-01383-f004:**
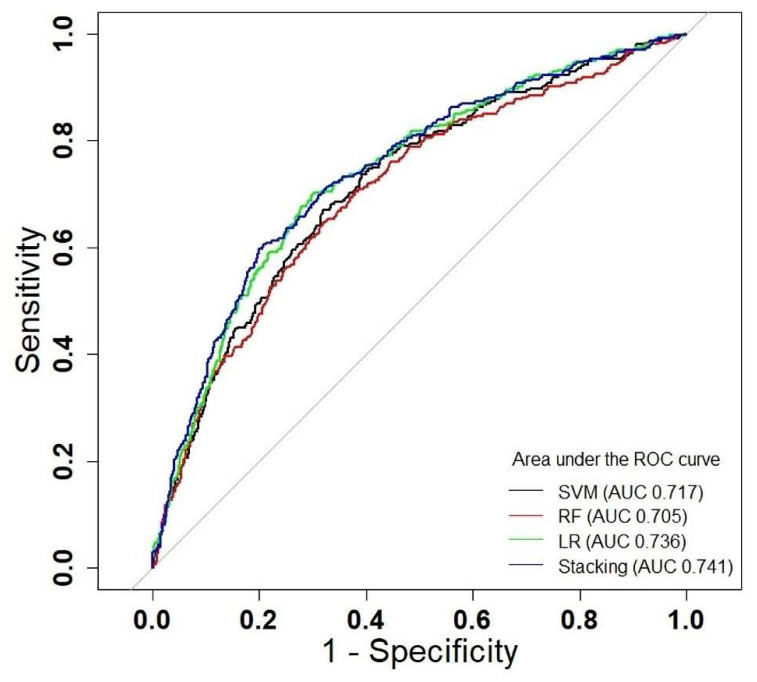
Evaluation of the predictive ability of non-fatal drowning prediction models based on the test dataset. Area under average ROC curve 0.741 (stacking), 0.736 (LR), 0.717 (SVM), and 0.705 (RF).

**Table 1 children-09-01383-t001:** Univariate analysis on the risk factors of drowning in children.

	Non-Fatal Drowning		χ^2^	*p*
	No (%)	Yes (%)		
overall	7307 (87.8)	1013 (12.2)		
Grade			82.987	<0.01
grade 3–6	4457 (61.0)	768 (75.81)		
grade 7–8	2850 (39.0)	245 (24.19)		
Gender			61.23	<0.01
Males	3720 (50.91)	649 (64.07)		
Females	3587 (49.09)	346 (35.93)		
Personality			32.37	<0.01
Introvert	1094 (14.97)	193 (19.05)		
Extrovert	3836 (50.50)	538 (53.11)		
Mild	1301 (17.80)	116 (11.45)		
Do not know	1076 (14.73)	166 (16.39)		
Relationships with classmates			58.092	<0.01
Very good	3642 (49.84)	467 (46.10)		
Good	3360 (45.98)	449 (44.32)		
Not good	182 (2.49)	53 (5.23)		
Bad	123 (1.68)	44 (4.34)		
Relationships with family members			80.381	<0.01
Very good	5158 (70.59)	645 (63.67)		
Good	1908 (26.11)	284 (28.04)		
Not good	187 (2.56)	51 (5.03)		
Bad	54 (0.74)	33 (3.26)		
Number of siblings			6.05	0.05
One	629 (8.61)	106 (10.46)		
Two	3341 (45.72)	478 (47.19)		
Three or over	3337 (45.67)	429 (42.35)		
Home ranking			5.80	0.06
First	2835 (38.80)	358 (35.34)		
Second	2422 (33.15)	370 (36.53)		
Third or over	2050 (28.05)	285 (28.13)		
Is open water near home or school well protected?			1.74	0.42
Yes	6091 (83.36)	859 (84.80)		
No	806 (11.03)	98 (9.67)		
No open water	410 (5.61)	56 (5.53)		
Would you like to swim in open water with a warning sign?			260.23	<0.01
Yes	229 (3.13)	80 (7.90)		
Probably	420 (5.75)	155 (15.30)		
Probably not	730 (9.99)	170 (16.78)		
Not	5928 (81.13)	608 (60.02)		
Distance between the school and the surrounding open waters (Meters)			13.75	0.008
<100	1553 (21.25)	259 (25.57)		
100–500	997 (13.65)	137 (13.52)		
500 +	1196 (16.37)	170 (16.78)		
Have no water area	1202 (16.45)	167 (16.49)		
Do not know	2359 (32.28)	280 (27.64)		
Distance from home to open water (Meters)			3.97	0.41
<100	1846 (25.26)	241 (23.79)		
100–500	1356 (18.56)	204 (20.14)		
500 +	1158 (15.85)	173 (17.08)		
Have no water area	1622 (22.20)	208 (20.53)		
Do not know	1325 (18.13)	187 (18.46)		
Swimming skill (Meters)			84.47	<0.01
≥100	786 (10.67)	189 (18.66)		
50–100	2101 (28.75)	308 (30.40)		
Over 500	1262 (17.27)	203 (20.04)		
Unable to swim	3158 (43.22)	313 (30.90)		
Frequency of swimming in open water			752	<0.01
≥three times per month	350 (4.79)	175 (17.28)		
Once or twice a month	337 (4.61)	150 (14.81)		
Once or twice a season	235 (3.22)	104 (10.27)		
Once or twice a year	284 (3.88)	117 (11.54)		
Zero	6101 (83.50)	467 (46.10)		

**Table 2 children-09-01383-t002:** Prediction performance of these models in the test set.

Outcome and Model	AUC	Sensitivity	F1 Value	Accuracy	Specificity
Logistic Regression	0.736	0.605	0.352	0.740	0.758
Random Forest	0.705	0.667	0.311	0.655	0.654
Support Vector Machine	0.717	0.581	0.331	0.726	0.745
Ensemble Learning	0.741	0.625	0.359	0.739	0.754

## Data Availability

The data used to support the findings of this study are available from the corresponding author upon request.

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
