# Peer review of "Non-Fatal Drowning Risk Prediction Based on Stacking Ensemble Algorithm"

_children, 2022, doi:10.3390/children9091383_

Round 1

Reviewer 1 Report

Thank you for submitting this manuscript.  Questions that may come up could include:

What risk factors have you identified that were not already identified?  What have you added that was not already understood?  How does your method increase the likelihood that action will be taken?

I ask these questions to make you aware of what will likely be a common sentiment: This is a fancy way of showing something obvious.  Responding to this perception and/or correcting this misperception will make your paper stronger.

Below are a few specific comments/suggestions:

line 43: Drowning is the leading cause of injury death in children aged 1–14 years[3]."  Please qualify more specifically - is this for the whole globe or more specifically for developing countries?  Also, the link for reference #3 does not work.

line 66: Consider defining "deep neural network" and how it is relevant to this work.

line 71: Consider pointing out that "imbalanced datasets" are defined in 2.3.1.

2.2. Data Collection: Consider explaining why you added personality characteristics to your survey.

Please proofread for English fluency.  This is good and can be better.

Author Response

Response to Reviewer 1 Comments

Dear Ms. Estelle Ding and professor Reviewer,

Thank you for your careful review of our paper and efficient decision-making process, and thank you very much for giving us an opportunity to revise our manuscript. We appreciate reviewer and editor very much for their positive and constructive comments and suggestions to our manuscript entitled “Non-fatal Drowning risk prediction Based on Stacking Ensemble algorithm”. We thoroughly reviewed the reviewers' comments and made revisions, which were highlighted in the revised manuscript. The details regarding the revisions are as follows.

What risk factors have you identified that were not already identified?

Response: Thanks for your time and consideration. The details regarding the revisions are as follows. Compared to previous studies, our study found that personality was an important risk factor and the introvert were more likely to suffer non-fatal drowning.

What have you added that was not already understood? 

Response: Thanks for your time and consideration. A few of articles using machine learning algorithms to predict the risk of non-fatal drowning. Therefore, we use several machine learning algorithms to develop different non-fatal drowning prediction models based on the risk factors. Compared with other classification algorithms, we found that stacking ensemble algorithms achieved better in non-fatal drowning prediction. Applying ML algorithms to drowning prediction may help guide children drowning prevention.

How does your method increase the likelihood that action will be taken?

Response: Thanks for your time and consideration. On the one hand, the local government can use this method to estimate the risk of children non-fatal drowning and identified the high-risk individuals. And the targeted prevention can be carried out and helps to decrease the occurrence of children drowning. On the other hand, we can prevent children from drowning by controlling for these correlated variables. When a child is found to have a high risk of drowning, we will control for risk factors by ranking the importance of the 9 variables. For example, our research shows that the frequency of swimming in open water is the most important variable. We can take some measures such as checking open water regular and putting up warning signs in open water.

Below are a few specific comments/suggestions:

line 43: Drowning is the leading cause of injury death in children aged

1–14 years[3]."  Please qualify more specifically - is this for the

whole globe or more specifically for developing countries?  Also, the

link for reference #3 does not work.

Response: Thanks for your time and consideration. This sentence is this for the whole globe. We have rephrased the sentence: Drowning is the leading cause of injury death in children aged 1–14 years around the world[3]. At the same time, I have made changes to the references: Organization WH. Global report on drowning: preventing a leading killer; WHO Press. Spain. 2014.( It's so strange that the link to Reference #3 of Reference #3 is still working in my computer.)

line 66: Consider defining "deep neural network" and how it is relevant

to this work.

Response: Thanks for your time and consideration. As a classical algorithm in machine learning, neural networks own strong predictive ability. Based on your suggestion, perhaps it would be more appropriate to remove the term. We have rephrased the sentence: A previous study in South Korea found the ML can improve the prediction of long-term outcomes in ischemic stroke patients[19].

line 71: Consider pointing out that "imbalanced datasets" are defined in 2.3.1.

Response: Thanks for your valuable comment. We have added the definition of imbalanced datasets in 2.3.1: Unbalanced dataset is types of data in which the number of observations in one or some subclasses is much larger than others[22].

2.2. Data Collection: Consider explaining why you added personality

characteristics to your survey."

Response: In most studies, personality is not included as drowning risk factor. However, some studies found that personality is also risk factor of drowning. At the same time, this variable is easy to obtain when doing a questionnaire. Finally, we have added personality characteristics to our survey.

We appreciate the reviewers for your constructive suggestions and providing us the opportunity to revise and improve our manuscript. The issues raised have been addressed. And we hope that our revision could answer your concerns and comments. Thank you again for your comments.

3     Organization WH. Global report on drowning: preventing a leading killer; WHO Press. Spain. 2014.

22     Searle, S.R. Linear Models for Unbalanced Data; Wiley Press, New York, 1987.

Reviewer 2 Report

The paper presents an interesting model about risk factors of drawing in children. No maijor concerns

Author Response

Response to Reviewer 2 Comments

Dear Ms. Estelle Ding and professor Reviewer,

Thank you for your careful review of our paper and efficient decision-making process, and thank you very much for giving us an opportunity to revise our manuscript. We appreciate reviewer and editor very much for their positive and constructive comments and suggestions to our manuscript entitled “Non-fatal Drowning risk prediction Based on Stacking Ensemble algorithm”. We thoroughly reviewed the reviewers' comments and made revisions, which were highlighted in the revised manuscript. The details regarding the revisions are as follows.

Comments and Suggestions for Authors

The paper presents an interesting model about risk factors of drawing in children. No major concerns

Authors’ response: We appreciate the reviewers for your constructive suggestions and providing us the opportunity to revise and improve our manuscript. Thank you again for your comments.

Reviewer 3 Report

This study compared the risk prediction ability of nonfatal drowning of three different machine learning algorithms and a developed a stacking ensemble model using survey data of students in a city in China.

Major point:

It is possible to find out the risk itself when individual data is input into the developed machine learning algorithm, but as far as I know, it is not possible to know if it is due to certain variables. Then, when a child is found to have a high risk of drowing, it is questionable how it can help to reduce the risk of drowning by correcting certain variables.

Minor point:

Introduction

Lines 39 and 43 are duplicates. Please delete one of them.

There is a duplicate sentence on lines 44-45. Please delete one of them.

In the introduction, ML first appears on line 59. It seems that the abbreviation that appears for the first time should be written out, and the abbreviation should be used from then on. I suggest modifying ML to Machine Learning (ML). It would be good to unify all machine learning that comes out later with ML.

In the introduction, drowning and nonfatal drowning are used interchangeably. Perhaps you analyzed the risks of nonfatal drowning rather than fatal drowning, I think, because you used survey data of living children. It would be nice if you could describe it.

Methods

Supplementary file

I do not understand the meaning of question 5 and its answers in the Drowning-related knowledge and behavior category. Please check if there are any errors in the translation.

The answer to question 7 in the Drowning-related knowledge and behavior category depends on whether the victim is an adult or a child. It is necessary to check whether the questions in the questionnaire are expressed in this way.

In the sub-question of question 9 of the Drowning-related knowledge and behavior category, question c asks about the pool, and the answer describes the pond. It is necessary to check whether the translation is an error.

Question 21 in the Drowning-related knowledge and behavior category seems to be incomplete. It is necessary to check whether the translation is an error.

It would be good to put the unit of currency in the answer to question 28 of the Drowning-related knowledge and behavior category.

Results

3.1. Univariate Variables Selection

The percentage of table 1 is expressed as the percentage of col. It seems to be easier to understand when expressed as a percentage by category (row) of variables for each group. For example, the percentage for each group of introvert of the personality variable should be expressed as 15% (1094/7307*100) for No and 19.1% (193/1013*100) for Yes.

The results described in the results session and the results described in the discussion session are different. In the results, it would be better to describe both the variables that contribute to the risk of nonfatal drowning and the variables that do not. And please correct the results session and discussion session so that the results do not conflict.

Line 181 states "The extroverts are more likely 181 to suffer non-fatal drowning than other students (χ2=32.37, P<0.001"), but introverts are described in the discussion. Modifications are required after confirmation.

There are some questions that are included in the questionnaire but not listed in Table 1, such as the question of going swimming alone without a guardian in Table 1. Even if the results are not statistically significant, it would be better to describe them in Table 1.

Discussion

Better swimming skills are identified as a risk factor for nonfatal drowning. Children are vulnerable to drowning due to lack of swimming skills, which is expressed in the introduction and discussion, but conflicting results. I think you should explain the reason for this result. In addition, there are contents that are different from those identified as risk factors in previous research results. You have to explain it.

I am happy to review an interesting study.

Author Response

Dear Ms. Estelle Ding and professor Reviewer,

Thank you for your careful review of our paper and efficient decision-making process, and thank you very much for giving us an opportunity to revise our manuscript. We appreciate reviewer and editor very much for their positive and constructive comments and suggestions to our manuscript entitled “Non-fatal Drowning risk prediction Based on Stacking Ensemble algorithm”. We thoroughly reviewed the reviewers' comments and made revisions, which were highlighted in the revised manuscript. The details regarding the revisions are as follows.

This study compared the risk prediction ability of nonfatal drowning of three different machine learning algorithms and a developed a stacking ensemble model using survey data of students in a city in China.

Major point:

It is possible to find out the risk itself when individual data is input into the developed machine learning algorithm, but as far as I know, it is not possible to know if it is due to certain variables. Then, when a child is found to have a high risk of drowning, it is questionable how it can help to reduce the risk of drowning by correcting certain variables.

Authors’ response: Thanks for your time and consideration. As you can see, it is impossible for us to know if it is due to certain variables. What we can know is that the correlation between these variables and drowning is very strong. Therefore, we can prevent most of children (not all children) from drowning by controlling for these strongly correlated variables. When a child is found to have a high risk of drowning, we will control for risk factors by ranking the importance of the 9 variables. For example, our research shows that the frequency of swimming in open water is the most important variable. We can take some measures such as checking open water regular and putting up warning signs in open water.

Minor point:

Introduction

Lines 39 and 43 are duplicates. Please delete one of them.

Authors’ response: Thanks for your time and consideration. Based on your suggestion, we have deleted lines39.

There is a duplicate sentence on lines 44-45. Please delete one of them.

Authors’ response: Thanks for your time and consideration. Based on your suggestion, we have deleted the duplicate sentence on lines 45.

In the introduction, ML first appears on line 59. It seems that the abbreviation that appears for the first time should be written out, and the abbreviation should be used from then on. I suggest modifying ML to Machine Learning (ML). It would be good to unify all machine learning that comes out later with ML.

Authors’ response: Thanks for your time and consideration. Based on your suggestion, we have completed all modification in original paper.

Line 59: However, previous studies mostly focused on the risk factors of non-fatal drowning, only a few studies paid attention to develop better Machine Learning (ML) model[15].

Line 65: A previous study in South Korea found ML can improve the prediction of long-term outcomes in ischemic stroke patients[19].

Line 67: Studies about ML algorithms in drowning prediction are rare, and most previous studies implemented single learning algorithms.

In the introduction, drowning and nonfatal drowning are used interchangeably. Perhaps you analyzed the risks of nonfatal drowning rather than fatal drowning, I think, because you used survey data of living children. It would be nice if you could describe it.

Authors’ response: Thanks for your valuable comment. We have added the definition of nonfatal drowning on line 76: Drowning is a process in which breathing is impaired by submersion/immersion in a liquid[20]. It includes non-fatal drowning and fatal drowning. If a person is rescued at any time, the process of drowning is interrupted, which is termed nonfatal drowning[21].

Methods

Supplementary file

I do not understand the meaning of question 5 and its answers in the Drowning-related knowledge and behavior category. Please check if there are any errors in the translation.

Authors’response: Thanks for your time and consideration. Drowning-related information included general information, drowning-related knowledge and behavior. Question 5 is general information. Question 5 refers to the relationship between children and their classmates. Some studies have shown that relationships with classmates can influence drowning behaviour. Children who have poor relationships with their classmates are more likely to drown. Based on your suggestion, we have rephrased the question 5: How is your relationship with your classmates?

The answer to question 7 in the Drowning-related knowledge and behavior category depends on whether the victim is an adult or a child. It is necessary to check whether the questions in the questionnaire are expressed in this way.

Authors’ response: Thanks for your time and consideration. Drowning-related information included general information, drowning-related knowledge and behavior. Question 7 is general information. Based on your suggestion, we have rephrased the question 7: How many siblings are in your family (including yourself)?

In the sub-question of question 9 of the Drowning-related knowledge and behavior category, question c asks about the pool, and the answer describes the pond. It is necessary to check whether the translation is an error.

Authors’ response: Thanks for your time and consideration. Based on your suggestion, we have rephrased the question c: Are all construction site pools emptied in a timely manner? ① Yes ② No ③ No pools

Question 21 in the Drowning-related knowledge and behavior category seems to be incomplete. It is necessary to check whether the translation is an error.

Authors’ response: Thanks for your time and consideration. Based on your suggestion, we have rephrased the question 21: Do you remember the month of this drowning?

It would be good to put the unit of currency in the answer to question 28 of the Drowning-related knowledge and behavior category.

Authors’ response: Thanks for your time and consideration. Based on your suggestion, we have rephrased the question 28: What was the cost of the drowning treatment?

① No cost ② <10¥ ③ 10-99¥ ④ 100-999¥ ⑤ 1000-2999¥ ⑥ 3000-9999¥ ⑦ 10000-29999¥ ⑧ ≥30000¥ ⑨ Don't know ⑩ No drowning

Results

3.1. Univariate Variables Selection

The percentage of table 1 is expressed as the percentage of col. It seems to be easier to understand when expressed as a percentage by category (row) of variables for each group. For example, the percentage for each group of introvert of the personality variable should be expressed as 15% (1094/7307*100) for No and 19.1% (193/1013*100) for Yes.

Authors’ response: Thanks for your time and consideration. Based on your comments, we have rephrased the table, the detailed results are as follows:

This univariate feature selection method uses the chi-squared (χ2) test for non-fatal drowning attributes to select 10 best variables our dataset. Table 1 describes the distribution of non-fatal drownings among different subgroups. A total of 8320 children with 4369 males and 3951 females were included in the study, about 12.2% (1013/8320) experienced non-fatal drowning during the past year, the drowning ratio of students in grades 3 to 6 (75.81%, 768/1013) and males (64.07%, 649/1013) who had ever experienced non-fatal drowning are higher than their counterparts grade 7-8 (24.19%,245/1013), female (35.93%,364/1013). The extroverts are more likely to suffer non-fatal drowning than other students (χ2=32.37, P<0.001). The students with better swimming skills are more likely to experience non-fatal drowning.

Table 1. Univariate analysis on the risk factors of drowning in children.

Non-fatal drowning

χ2

P

No (%)

Yes (%)

overall

7307 (87.8)

1013 (12.2)

Grade

82.99

<0.01

grade 3-6

4457(61.0)

768(75.81)

grade 7-8

2850(39.0)

245(24.19)

Gender

61.23

<0.01

Males

3720(50.91

649(64.07)

Females

3587(49.09)

364(35.93)

Personality

32.37

<0.01

Introvert

1094(14.97)

193(19.05)

Extrovert

3836(52.50)

538(53.11)

Mild

1301(17.80)

116(11.45)

Do not know

1076(14.73)

166(16.39)

Relationships with classmates

58.09

<0.01

Very good

3642(49.84)

467(46.10)

Good

3360(45.98)

449(44.32)

Not good

182(2.49)

53(5.23)

Bad

123(1.68)

44(4.34)

Relationships with family members

80.38

<0.01

Very good

5158(70.59)

645(63.67)

Good

1908(26.11)

284(28.04)

Not good

187(2.56)

51(5.03)

Bad

54(0.74)

33(3.26)

Number of siblings(a7)

6.05

0.05

One

629(8.61)

106(10.46)

Two

3341(45.72)

478(47.19)

Three or over

3337(45.67)

429(42.35)

Home ranking(a8)

5.80

0.06

Frist

2835(38.80)

358(35.34)

second

2422(33.15)

370(36.53)

Third or over

2050(28.05)

285(28.13)

Is open water near home or school well protected?

1.74

0.42

Yes

6091(83.36)

859(84.80)

No

806(11.03)

98(9.67)

No open water

410(5.61)

56(5.53)

Would you like to swim in open water with a warning sign?

260.23

<0.01

Yes

229(3.13)

80(7.90)

Probably

420(5.75)

155(15.30)

Probably not

730(9.99)

170(16.78)

Not

5928(81.13)

608(60.02)

Distance between the school and the surrounding open waters (Meters)

13.75

0.008

< 100

1553(21.25)

259(25.57)

100–500

997(13.65)

137(13.52)

500 +

1196(16.37)

170(16.78)

Have no water area

1202(16.45)

167(16.49)

Do not know

2359(32.28)

280(27.64)

Distance from home to open water (Meters)

3.97

0.41

< 100

1846(25.26)

241(23.79)

100–500

1356(18.56)

204(20.14)

 500 +

1158(15.85)

173(17.08)

Have no water area

1622(22.20)

208(20.53)

Do not know

1325(18.13)

187(18.46)

Swimming skill (Meters)

84.47

<0.01

≥100

786(10.76)

189(18.66)

50–100

2101(28.75)

308(30.40)

Over 500

1262(17.27)

203(20.04)

Unable to swim

3158(43.22)

313(30.90)

Frequency of swimming in open water

752.00

<0.01

≥ three times per month

350(4.79)

175(17.28)

Once or twice a month

337(4.61)

150(14.81)

Once or twice a season

235(3.22)

104(10.27)

Once or twice a year

284(3.88)

117(11.54)

Zero

6101(83.50)

467(46.10)

The results described in the results session and the results described in the discussion session are different. In the results, it would be better to describe both the variables that contribute to the risk of nonfatal drowning and the variables that do not. And please correct the results session and discussion session so that the results do not conflict. Line 181 states "The extroverts are more likely 181 to suffer non-fatal drowning than other students (χ2=32.37, P<0.001"), but introverts are described in the discussion. Modifications are required after confirmation.

Authors’ response: Thanks for your time and consideration. Based on your suggestion, we have modified the sentence line 221 and line 294: In this study, we found that frequency of swimming in open water, distance from open water around school, swimming skills(unable to swim), personality(extrovert) and ect. are most closely associated with children drowning. Our study shows frequency of swimming in the open water, distance between school and surrounding open waters, swimming skills (unable to swim), personality(extrovert), relationship with family members, relationship with classmates, distance between the school and the surrounding open waters, sex and grade are closely associated with children non-fatal drowning.

At the same time, we have described both the variables that contribute to the risk of non-fatal drowning and the variables that do not: We removed variables that are not statistically significant. For examples: number of siblings, family ranking, whether open water is well protected near the home or school, and distance from home to open water.

There are some questions that are included in the questionnaire but not listed in Table 1, such as the question of going swimming alone without a guardian in Table 1. Even if the results are not statistically significant, it would be better to describe them in Table 1.

Authors’ response: Thanks for your time and consideration. Based on your suggestion, we have added variables that are not statistically significant in Table 1. There are some questions that are included in the questionnaire but not listed in Table 1. The details reasons are as follows.

① We removed age due to the fact that it was linked to grade and our survey is conducted by grade.

② As they are all drowning-related risk behaviours: frequency of swimming in open water unaccompanied by an adult in the past 12 months, frequency of fishing alone in the past 12 months, frequency of playing at a pond in the past 12 months, and frequency of diving or jumping in open water at unknown depths in the past 12 months. We have combined them into one question: frequency of swimming in open water (in the past 12 months).

③ As some of the questions have the same meaning, we have kept only one of them. (“Distance from home to open water (Meters)” and “Are there open water areas on the way to school?”. We have retained the first.).

④Of the five sub-questions in question 9 in the category of knowledge and behaviour related to drowning. We have combined them into one question: Is open water near home or school well protected?

Discussion

Better swimming skills are identified as a risk factor for nonfatal drowning. Children are vulnerable to drowning due to lack of swimming skills, which is expressed in the introduction and discussion, but conflicting results. I think you should explain the reason for this result. In addition, there are contents that are different from those identified as risk factors in previous research results. You have to explain it.

Authors’ response: Thanks for your time and consideration. Based on your comments, we have rephrased the sentence:

Abstract line 26: We found risk factors including frequency of swimming in the open water, distance between the school and the surrounding open waters, swimming skills(unable to swim), personality(introvert) and relationality with family members were closely associated with non-fatal drowning. (abstract)

Result line 182: The students without swimming skills are more likely to experience non-fatal drowning.

Result line 187: The critical 9 factors were frequency of swimming in the open water, distance between school and surrounding open waters, swimming skills (unable to swim), personality(introvert), relationship with family members, relationship with classmates, distance between the school and the surrounding open waters, sex and grade.

Discuss line 221: In this study, we found that frequency of swimming in open water, distance from open water around school, swimming skills (unable to swim), personality (introvert) and ect.

Conclusion line 294: Our study shows frequency of swimming in the open water, distance between school and surrounding open waters, swimming skills (unable to swim), personality(introvert), relationship with family members, relationship with classmates, distance between the school and the surrounding open waters, sex and grade are closely associated with children non-fatal drowning.

We appreciate the reviewers for your constructive suggestions and providing us the opportunity to revise and improve our manuscript. The issues raised have been addressed. And we hope that our revision could answer your concerns and comments. Thank you again for your comments.

References

  1. van Beeck, E.F.; Branche, C.M.; Szpilman, D.; Modell, J.H.; Bierens, J.J. A new definition of drowning: towards documentation and prevention of a global public health problem. Bull World Health Organ 2005, 83, 853-856.
  2. Szpilman, D.; Bierens, J.J.; Handley, A.J.; Orlowski, J.P. Drowning. N Engl J Med 2012, 366, 2102-2110, doi:10.1056/NEJMra1013317.

Round 2

Reviewer 1 Report

Please proofread for flow.  The English is quite good, but be sure to read it out loud as there are several awkward glitches.

Author Response

Dear Ms. Estelle Ding and professor Reviewer,

Thank you for your careful review of our paper and efficient decision-making process, and thank you very much for giving us an opportunity to revise our manuscript. Our deepest gratitude goes to you for your careful work and thoughtful suggestions that have helped improve this paper substantially. We thoroughly reviewed the reviewers' comments and made revisions, which were highlighted in the revised manuscript. Revised portion are marked in red in the paper. The details regarding the revisions are as follows.

Comments and Suggestions for Authors

Please proofread for flow. The English is quite good, but be sure to read it out loud as there are several awkward glitches.

Response:We apologize for the language problems in the original manuscript. Language presentation was improved with assistance from a native English speaker with appropriate research background. We really hope that the flow and language level have been substantially improved. Thank you very much for your patience and explanation. Those comments are all valuable and very helpful for revising and improving our paper, as well as the important guiding significance to our researches. Thank you once again for your attention to our paper.

Reviewer 3 Report

Thank you for your hard work in editing the thesis according to the review opinion.

However, there are some incorrect corrections, and there are some inconsistencies in the results, so it needs to be reviewed.

Author Response

Dear Ms. Estelle Ding and professor Reviewer,

Thank you for your careful review of our paper and efficient decision-making process, and thank you very much for giving us an opportunity to revise our manuscript. Our deepest gratitude goes to you for your careful work and thoughtful suggestions that have helped improve this paper substantially. We thoroughly reviewed the reviewers' comments and made revisions, which were highlighted in the revised manuscript. Revised portion are marked in red in the paper. The details regarding the revisions are as follows.

Methods

Supplementary file

I do not understand the meaning of question 5 and its answers in the Drowning-related knowledge and behavior category. Please check if there are any errors in the translation.

Authors’response: Thanks for your time and consideration. Drowning-related information included general information, drowning-related knowledge and behavior. Question 5 is general information. Question 5 refers to the relationship between children and their classmates. Some studies have shown that relationships with classmates can influence drowning behaviour. Children who have poor relationships with their classmates are more likely to drown. Based on your suggestion, we have rephrased the question 5: How is your relationship with your classmates?

è The question I commented on was the following

  1. 2 minutes after droning, you will lose consciousness and suffer irreversible damage to your nervous system.

4 to 6 7 to 10 11 to 15 Don't know

Authors’ response: We apologize for the language problems in the supplementary file. Sorry to have taken up your time once again. Based on your suggestion, we have rephrased the question 5: Two minutes after drowning, people will be unconscious. How many minutes does the nervous system suffer irreversible damage? ① 4 to 6 ② 7 to 10 ③ 11 to 15 ④ Don't know

The answer to question 7 in the Drowning-related knowledge and behavior category depends on whether the victim is an adult or a child. It is necessary to check whether the questions in the questionnaire are expressed in this way.

Authors’ response: Thanks for your time and consideration. Drowning-related information included general information, drowning-related knowledge and behavior. Question 7 is general information. Based on your suggestion, we have rephrased the question 7: How many siblings are in your family (including yourself)?

è The question I commented on was the following

  1. In CPR, the ratio of the number of heart compressions to the number of artificial breaths is

5:1   15:2   30:2   Don't know

Authors’ response: We apologize for the language problems in the supplementary file. We are sorry that we did forget that the ratios of adult CPR (30:2) and children CPR (15:2) are different. Based on your suggestion, we have rephrased the question 7: What is the ratio of the frequency of cardiac compressions to the number of artificial breaths during CPR in children? ① 5:1   ② 15:2   ③ 30:2   ④ Don't know

Results

3.1. Univariate Variables Selection

The percentage of table 1 is expressed as the percentage of col. It seems to be easier to understand when expressed as a percentage by category (row) of variables for each group. For example, the percentage for each group of introvert of the personality variable should be expressed as 15% (1094/7307*100) for No and 19.1% (193/1013*100) for Yes.

Authors’ response: Thanks for your time and consideration. Based on your comments, we have rephrased the table, the detailed results are as follows:

This univariate feature selection method uses the chi-squared (χ2) test for non-fatal drowning attributes to select 10 best variables our dataset. Table 1 describes the distribution of non-fatal drownings among different subgroups. A total of 8320 children with 4369 males and 3951 females were included in the study, about 12.2% (1013/8320) experienced non-fatal drowning during the past year, the drowning ratio of students in grades 3 to 6 (75.81%, 768/1013) and males (64.07%, 649/1013) who had ever experienced non-fatal drowning are higher than their counterparts grade 7-8 (24.19%,245/1013), female (35.93%,364/1013). The extroverts are more likely to suffer non-fatal drowning than other students (χ2=32.37, P<0.001). The students with better swimming skills are more likely to experience non-fatal drowning.

è Changes in table 1 are not reflected in the pdf version.

Authors’ response: We are extremely grateful to reviewer for pointing out this problem. Based on your suggestion, we have rephrased the pdf version.

The results described in the results session and the results described in the discussion session are different. In the results, it would be better to describe both the variables that contribute to the risk of nonfatal drowning and the variables that do not. And please correct the results session and discussion session so that the results do not conflict.

Line 181 states "The extroverts are more likely 181 to suffer non-fatal drowning than other students (χ2=32.37, P<0.001"), but introverts are described in the discussion. Modifications are required after confirmation.

Authors’ response: Thanks for your time and consideration. Based on your suggestion, we have modified the sentence line 221 and line 294: In this study, we found that frequency of swimming in open water, distance from open water around school, swimming skills(unable to swim), personality(extrovert) and ect. are most closely associated with children drowning. Our study shows frequency of swimming in the open water, distance between school and surrounding open waters, swimming skills (unable to swim), personality(extrovert), relationship with family members, relationship with classmates, distance between the school and the surrounding open waters, sex and grade are closely associated with children non-fatal drowning.

At the same time, we have described both the variables that contribute to the risk of non-fatal drowning and the variables that do not: We removed variables that are not statistically significant. For examples: number of siblings, family ranking, whether open water is well protected near the home or school, and distance from home to open water.

è Looking at the results in Table 1, the lower the grade, the men, the more introspective (actually, the proportion of people who answered mild and do not know is difficult to interpret), the worse the relationship with classmates and family members, the less sibling, the higher willingness to swim in open water, the shorter distance from the school to open water, the better swimming skills, and the more frequent swimming in open water was associated the high risk of non-fatal drowning. This result and the variables from the random forest should have the same direction. However, the author describes swimming skills as "unable to swim" and personality as "introvert", which is different from the results of logistic regression. If it is not that the results of the multivariate logistic regression were different from the results of the univariate logistic regression, it seems that the results should be described by reviewing the random forest results again. After reviewing the results again, the contents of the discussion also need to be corrected.

Authors’ response: Thank you for your careful review. Thank you for your precious comments and advice. Based on your suggestion, we have rephrased the sentences from ② to ⑤. We have added content to the discussion section. The details revisions are as follows.

①Sibling rankings were not included in our model because the variables in our model all had p-values less than 0.01.

②Result line 204 and 205: The introverts are found to be more likely to suffer non-fatal drowning than other students (χ2=32.37, P<0.01). The students with self-reported better swimming skills are also more likely to experience non-fatal drowning.

③Result line 210: The 9 critical factors were frequency of swimming in open water, distance between school and surrounding open waters, swimming skills, personality, relationship with family members, relationship with classmates, distance between the school and the surrounding open waters, sex and grade.

④Discussion line 247: In this study, we found that frequency of swimming in open water, distance from open water around school, swimming skills, personality (Introvert) and etc. are most closely correlated with children drowning.

⑤Conclusion line 298: Our study shows that the frequency of swimming in open water, the distance between school and surrounding open waters, swimming skills, personality(introvert), relationship with family members, relationship with classmates, distance between the school and the surrounding open waters, sex and grade are closely correlated with children experiencing non-fatal drowning.

⑥Added in discussion line 256: Self-reported swimming skills were admittedly shown to be associated with drowning in this study. In rural areas, children with better swimming skills might swim more frequently and engage in risk-taking behaviors while in the water, which may have contributed to our results. There is no doubt that poor swimmers are certainly more likely to drown when wading than those who swim well.

Thank you very much for your patience and explanation. Those comments are all valuable and very helpful for revising and improving our paper, as well as the important guiding significance to our researches. Thank you once again for your attention to our paper.

Round 3

Reviewer 3 Report

Thank you for your careful revision according to the content of your review.